# The Molecular Basis for Selectivity of the Cytotoxic Response of Lung Adenocarcinoma Cells to Cold Atmospheric Plasma

**DOI:** 10.3390/biom13111672

**Published:** 2023-11-20

**Authors:** Mikhail Biryukov, Dmitriy Semenov, Nadezhda Kryachkova, Alina Polyakova, Ekaterina Patrakova, Olga Troitskaya, Elena Milakhina, Julia Poletaeva, Pavel Gugin, Elena Ryabchikova, Dmitriy Zakrevsky, Irina Schweigert, Olga Koval

**Affiliations:** 1Institute of Chemical Biology and Fundamental Medicine, Siberian Branch of the Russian Academy of Sciences, 630090 Novosibirsk, Russia; biryukov.mm@ya.ru (M.B.); semenov@niboch.nsc.ru (D.S.); krychkovanv@rambler.ru (N.K.); a.polyakova1@g.nsu.ru (A.P.); sennuie@gmail.com (E.P.); troitskaya_olga@bk.ru (O.T.); fabaceae@yandex.ru (J.P.); lenryab@niboch.nsc.ru (E.R.); 2Department of Natural Sciences, Novosibirsk State University, 630090 Novosibirsk, Russia; 3Khristianovich Institute of Theoretical and Applied Mechanics, Siberian Branch of the Russian Academy of Sciences, 630090 Novosibirsk, Russia; lena.yelak@gmail.com (E.M.); guginpavel@gmail.com (P.G.); zakrdm@gmail.com (D.Z.); ivschweigert@gmail.com (I.S.); 4Department of Radio Engineering and Electronics, Novosibirsk State Technical University, 630073 Novosibirsk, Russia

**Keywords:** cold atmospheric plasma, transcriptome analysis, cell-cycle arrest, GADD45, p53 pathway, endoplasmic reticulum stress, GPx4

## Abstract

The interaction of cold atmospheric plasma (CAP) with biotargets is accompanied by chemical reactions on their surfaces and insides, and it has great potential as an anticancer approach. This study discovers the molecular mechanisms that may explain the selective death of tumor cells under CAP exposure. To reach this goal, the transcriptional response to CAP treatment was analyzed in A549 lung adenocarcinoma cells and in lung-fibroblast Wi-38 cells. We found that the CAP treatment induced the common trend of response from A549 and Wi-38 cells—the p53 pathway, KRAS signaling, UV response, TNF-alpha signaling, and apoptosis-related processes were up-regulated in both cell lines. However, the amplitude of the response to CAP was more variable in the A549 cells. The CAP-dependent death of A549 cells was accompanied by DNA damage, cell-cycle arrest in G2/M, and the dysfunctional response of glutathione peroxidase 4 (GPx4). The activation of the genes of endoplasmic reticulum stress and ER lumens was detected only in the A549 cells. Transmission-electron microscopy confirmed the alteration of the morphology of the ER lumens in the A549 cells after the CAP exposure. It can be concluded that the responses to nuclear stress and ER stress constitute the main differences in the sensitivity of tumor and healthy cells to CAP exposure.

## 1. Introduction

The search for new antitumor approaches is not limited to the development of new drugs, but also stimulates the application of advanced physical modalities. Cold atmospheric plasma (CAP) has been applied to kill various tumor cells [1,2,3,4,5,6,7,8,9] and spheroids [10,11] in vitro. It is a new and rapidly developed method of therapy for malignant neoplasms [12,13]. The CAP jet is a sequence of streamers that are generated in inert gases in the dielectric channel of a plasma-jet device and propagate along the gas stream in ambient air. It is formed at atmospheric pressure, and under certain conditions, its temperature does not exceed 39 °C, making it compatible with living bodies [14]. The anticancer effects of CAP rely on reactive oxygen species (ROS) and nitrogen (RNS), which are formed in the plasma jet [15,16]. These reactive species interact with biotargets to form secondary and, further, highly reactive molecules, changing the redox balance in the cell [17,18,19]. 

The ROS and RNS are formed in the cells in ordinary metabolic processes, and can regulate various cellular pathways. Under physiological conditions, superoxide anion radical (O_2_^•−^) and hydrogen peroxide (H_2_O_2_) act like key ROS signaling agents generated by the mitochondrial electron-transport chain and many enzymes that are regulated by growth factors and cytokines [20,21]. High intracellular concentrations of ROS cause oxidative stress. Oxidative stress is recognized by special systems based on nuclear-erythroid-related factor 2 (NRF2) and kelch-like ECH-associated protein 1 (KEAP1) [20]. Upon oxidation, KEAP1 facilitates the translocation of NRF2, allowing the nucleus to activate the expression of antioxidant genes [22]. The cellular capacity to overcome oxidative stress is different for rapidly divided tumor cells and healthy cells, because their basal metabolic rates are dissimilar. Tumor cells typically demonstrate a high level of free radicals compared to normal cells [23]. To detoxify ROS and escape apoptosis, tumor cells develop a protective mechanism by expressing elevated levels of antioxidant proteins and activating glycolysis to compensate the energy supply. Excessive exogenous ROS/RNS in cancer cells impairs their catabolism and can destabilize the energy pathways and stimulate cell death. It is known that selenium-containing glutathione peroxidases (GPxs)1-4 protect against oxidative challenge and inhibit inflammation and oxidant-induced regulated cell death [24]. Cysteine-containing GPX7 and GPX8 are unique in contributing to oxidative protein folding in the endoplasmic reticulum by reacting with protein isomerase as an alternate substrate. It is possible that glutathione peroxidases may also contribute to selective cell death under CAP exposure.

The application of the transcriptome analysis of differentially expressed genes (DEGs) in CAP-exposed cells was used to identify the major molecular pathways of CAP-dependent cell death, which most studies confirm as the p53 apoptotic signaling pathway, MAPK pathway and cell cycle progression [6,25,26,27]. At the same time, individual studies suggest the involvement of other pathways—necroptosis and autophagy [6], hypoxia [27], and metabolic processes [25]. Thus, the fine mechanism underlying the divergent cellular responses to CAP irradiation are not well understood. Previously, we experimentally determined the optimal discharge regime (or semi-selective regime) for the direct CAP jet treatment, under which lung adenocarcinoma A549, A427, and NCI-H23 cells demonstrated the substantial suppression of viability, coupled with a weak viability decrease in healthy lung-fibroblast Wi-38 and MRC-5 cells. Using the pan-caspase inhibitor, Z-VAD, we revealed that caspase-dependent cell-death pathways made a major contribution to CAP-dependent cell killing [3]. The study of the CAP-dependent activation of the autophagic cascade in exposed cells failed to explain the selectivity of caspase-dependent tumor-cell death [3]. 

Here, to determine the specific cellular response to the CAP treatment leading to cell survival or cell death, first, we compared the transcriptional changes in A549 and Wi-38 cells. The characteristics of the selected differentially expressed genes and their transcription factors made it possible to construct a map of interactions. Further, the identified changes were confirmed by independent molecular methods—PCR, Western blot, cell-cycle analysis, and electron microscopy.

## 2. Materials and Methods

### 2.1. Cell Cultures

The A549 human lung carcinoma cells (purchased: ATCC # CCL-185), Wi-38 (purchased: ATCC # CCL-75), and MRC-5 lung fibroblasts (ATCC # CCL-171, from Flow Laboratories, Oldham, UK) were grown in DMEM: Nutrient Mixture F-12 (DMEM:F12, Sigma-Aldrich, St. Louis, MO, USA) supplemented with 10% fetal bovine serum (GIBCO, Thermo Fisher Scientific, Waltham, MA, USA), 2 mM L glutamine, 250 mg/mL amphotericin B, and 100 U/mL penicillin/streptomycin in 5% CO_2_. The NCI-H23 epithelial-like lung adenocarcinoma (ATCC # CRL-5800) was grown in RPMI 1640 (GIBCO, Thermo Fisher Scientific, Waltham, MA, USA) supplemented with 10% fetal bovine serum (Sigma-Aldrich, St. Louis, MO, USA), 2 mM L-glutamine, 250 mg/mL amphotericin B and 100 U/mL penicillin/streptomycin.

### 2.2. Experimental Plasma Jet Setup

In our experiments, the source of the plasma jet was a discharge cell in the form of a coaxial dielectric channel with L = 100 mm and an inner diameter of 8 mm, and a capillary nozzle with L = 5 mm and diameter of 2.3 mm [2,4,28]. The discharge zone in the dielectric tube was formed by the powered electrode inside and the grounded ring outside of the tube (Figure 1a).

An ohmic high-impedance divider was used to measure the voltage *U*. Current measurements were carried out by a special sensor located perpendicular to the axis of CAP propagation and representing a collector—a flat metal electrode. Grounding of the collector through a low inductive resistance allowed us to register the frequency and amplitude of the current pulse *I*, reaching the collector. Irradiated objects were placed on the grounded metal collector, which allowed us to increase the electric field strength at the object surface and the rate of generation of active radicals in the contact zone of the jet with the object of exposure [4,29].

Culturing plates were placed so that cells were 25 mm from the nozzle. The fluid level above the cells was 3 mm. The plasma jet was generated in the laminar flow of helium, which was pumped through the dielectric channel of the plasma device. The typical conditions for CAP irradiation of the cells were as follows: sinusoidal voltage with the amplitude *U* = 3.3–3.5 kV (Figure 1b), the voltage frequency *f_U_* ≈ 52 kHz, and the helium-pumping rate υ = 9 L/min. These conditions of plasma-jet initiation provided a frequency of current touching the object of impact that was four times less than the *f_U_* frequency.

The spectrometer “Kolibri-2” [30] tuned to the wavelength range *λ* = 190–750 nm with an optical resolution of 0.17 nm was used for spectral measurements. The optical radiation in the region of contact between the plasma jet and the target was recorded at a distance of 3 mm at an angle of 30° degrees from the vertical and transmitted to the spectrometer through a multimode quartz light guide. Figure 1c shows typical luminescence spectra of the interaction region of the plasma jet with culture media. A set of spectral lines corresponding to helium He lines, molecular nitrogen N_2_ and nitrogen-ion N_2_^+^ lines, nitrogen oxide NO, hydroxide OH, and the Balmer series of hydrogen H_2_ lines, as well as O_2_ and O_2_^+^ lines, were observed. The inset in Figure 1c details the spectral line of OH hydroxide (*λ* ≈ 309 nm, A^2^Σ-X^2^Π transition).

All experiments were accompanied by temperature measurements of the plasma-jet–object interaction region using a Testo 872 thermal imager (TESTO, Hamburg, Germany) with a measurement accuracy of ±0.1 °C. The therapeutic application of plasma exposure is limited by the permissible heating of the object, which reduces the possible range of plasma-jet parameters. Therefore, in experiments on the plasma-jet exposure, attention was paid only to the final result of exposure—the temperature of the object and the conditions under which the temperature was acceptable for living objects were investigated. In the present work, the determination of the jet temperature seems irrelevant.

### 2.3. Cell-Viability Assay

Cell viability was detected 24 h after CAP irradiation or drug treatment using an 3-[4,5-dimethylthiazole-2-yl]-2,5-diphenyltetrazolium bromide (MTT test), as described previously [31].

### 2.4. Western Blot

Cells were lysed with cell-lysis buffer (50 mM Tris, pH 8.0, 5 mM EDTA, 150 mM NaCl) containing 0.1% SDS, 1x complete protease-inhibitor cocktail (Roche Diagnostics GmbH, Mannheim, Germany) and 1 mM PMSF. Western blot analysis of cell lysates was performed as described previously [32] using the following antibodies: anti-GPX4 (ab125066); anti-GPX7 (ab96257); anti-ATF3 (ab233797), anti-GADD45B (ab205252), γH2AFX (ab26350) and MDM2 (ab16895) (Abcam, Cambridge, UK), anti-β-Tubulin (T8328, Sigma-Aldrich, St. Louis, MO, USA); and goat anti-rabbit IgG (H+L) HRP-conjugated secondary antibodies (G-21234) and goat anti-mouse IgG (H+L) HRP-conjugated secondary antibodies (G-21040), both from (Invitrogen, Waltham, MA, USA).

### 2.5. Cell-Cycle Analysis

For analysis of cell-cycle phase distribution, at least 5 × 10^5^ cells were detached with trypsin and harvested with centrifugation at 100× *g* for 5 min 24 h after CAP treatment. Cells were washed with ice-cold PBS and fixed with ice-cold 70% methanol. Subsequently, the cells were washed with PBS and re-suspended in PBS containing 100 µg/mL RNase A (Magen Biotechnology, Guangzhou, China). After 5 min incubation, 50 µg/mL propidium iodide (Thermo Fisher Scientific, USA) solution in PBS was added for 15–20 min. Next, cell-cycle-distribution assessment was performed using FACSCanto II flow cytometer (BD Biosciences, Franklin Lakes, NJ, USA).

### 2.6. Measurement of Extracellular Nitrite Ions

Cells were treated with CAP for 1 or 2 min in 96-well plates. After 0–4 h, 50 µL aliquots of medium were placed in a new plate and concentration of nitrite ions was determined using Griess Reagent System (Promega, Madison, WI, USA). Nitrite standard was dissolved in full culture medium.

### 2.7. Transmission-Electron Microscopy

Cells were prepared for analysis as described previously [3]. Cells were collected by centrifugation. Cell pellets were resuspended in 4% paraformaldehyde, after which they were additionally fixed with 1% osmium-tetroxide solution, and sequential dehydration block sections were performed. Contrasted ultrathin sections were examined in a transmission-electron microscope JEM 1400, JEOL (Tokyo, Japan) equipped with a digital camera, Vleta, EM SIS (Muenster, Germany).

### 2.8. RNA Extraction and Real-Time Reverse-Transcription Polymerase Chain Reaction (RT-PCR)

Cells were lysed with the Lira reagent (Biolabmix, Novosibirsk, Russia) for 10 min at room temperature. Total RNA was extracted using phenol-chloroform method combined with silica-membrane purification according to manufacturer’s protocol. The RNA purity and concentration were assessed using Nanodrop spectrophotometer (Thermo Fisher Scientific, USA). To remove DNA contaminants, RNA samples were treated with DNase I (Thermo Fisher Scientific, Vilnius, Lithuania) according to manufacturer’s protocol.

The RT-PCR was performed with BioMaster RT-PCR SYBR Blue mix (Biolabmix, Novosibirsk, Russia) using LightCycler 96 System (Roche, Switzerland). A total of 40 ng of DNA-free RNA was used for the reaction. Relative mRNA levels were calculated using LightCycler 96 software as relative values normalized to the levels of *GAPDH* and *HPRT1* mRNAs. Primer efficiencies were determined with web-based LinRegPCR (https://www.gear-genomics.com, accessed on 19 October 2023) [33]. Gene-specific primers are summarized in Table 1. The levels of mRNAs are represented as relative values normalized to the level of *HPRT1* and *GAPDH* mRNA.

### 2.9. RNA-Seq and Transcriptome Analyses and Gene-Set-Enrichment Analysis

Cells were lysed by Trizol™ (Life Technologies, Carlsbad, CA, USA) according to the manufacturer’s protocol. Each sample was duplicated. The cDNA libraries were constructed according to standard Illumina recommendations. The sequencing was performed on the Illumina 1500 NextSeq platform at the Interdisciplinary Center of Collective Use of the Institute of Fundamental Medicine and Biology of Kazan (Volga Region) Federal University (Kazan, Russia). Bioinformatic analysis of NGS data was performed using the following software: Trimmomatic version v0.32 [34] to remove adapter sequences; bowtie2 to filter rRNA, tRNA and snRNA, SINE-, LINE-, and DNA-repeat consensus sequences; RepBase [35] (accessed 14 January 2022; Genetic Information Research Institute, Mountain View, CA, USA) for low-complexity simple repeats, as well as mitochondrial DNA (NC_012920.1); STAR 2.7.1a (ENCODE, Stanford University, Stanford, CA, USA) [36] to align experimental sequences with human genome/transcriptome assembly GRCh37/hg19 (NCBI RefSeq); QoRTs v1.3.6 [37] to analyze alignment quality and quantify individual RNA contributions; and DESeq2 1.36.0 [38] (R version 4.1.3 and Bioconductor 3.14) for differential analysis of gene expression. Following the differential gene-expression analysis, lists of up-regulated and down-regulated genes were analyzed with Enrichr using the R interface [39].

### 2.10. Statistical Analysis

Data were tested for normality, and if they followed a normal distribution, the mean  ±  standard deviation standard is presented, and Student’s *t* test was performed for comparisons between groups. A two-sided *p* value  <  0.05 was considered statistically significant.

## 3. Results

### 3.1. Transcriptome Analysis of A549 and Wi-38 Cells Treated with CAP

In order to determine the specific transcriptional changes in A549 cancer cells and healthy Wi-38 cells following CAP exposure, we analyzed the whole transcriptomes of these cells. Cells were exposed to CAP for 1 min and further incubated for 3 h or 24 h. Later, cells were lysed for total RNA extraction. The helium plasma jet was initiated with a generator of sinusoidal voltage with the amplitude of 3.3 kV at a fixed frequency f_U_ = 52 kHz. The samples of the total RNAs (Appendix A) were used for the whole-transcriptome analysis, as described in Methods. The differentially expressed genes (DEGs) in these two cell lines were subjected to a comparison. The relationships between individual transcriptomes and the groups of the transcriptomes of the Wi-38 and A549 cells, treated with CAP, indicate that the cells clustered into two different groups, forming two individual branches of the tree (Figure 2a). The changes that occurred as a result of the CAP exposure did not modify the structure of the tree. These findings indicate a global difference between the cell lines. It was demonstrated that the CAP-induced changes were significantly different for the A549 and Wi-38 cells (Figure 2b). These differences were particularly noticeable for the samples 24 h after the CAP treatment, and these results suggest a genuinely different molecular response to CAP under semi-selective conditions in these two cell lines. The common response trend identified in the coordinates used and depicted by the arrow (Figure 2b) was the same for both cell lines.

Next, the DEG sets of up-regulated genes and down-regulated genes were filtered as described in the Methods. Groups of 250 genes with the largest change for up- and down- regulation for each cell line were taken into the subsequent analysis. We constructed Venn diagrams and identified the number of genes with similar changes between cell lines and between time points within each individual cell line (Figure 3a). The number of common up-regulated genes for both cell lines for the two time points was six, and common down-regulated number was three, indicating a differential response to CAP exposure. The heatmap demonstrates the 50 most variable up-regulated and down-regulated differentially expressed genes in the Wi-38 and A549 cells (Figure 3b). It can be seen that these gene lists differ between the cell lines.

The gene-set-enrichment analysis for the top 250 transcripts that are up-regulated or down-regulated in the samples of A549 (Table 2) and Wi-38 (Table 3) cells (for 3 h and 24 h after CAP treatment) was applied. This analysis identified common transcription factors regulating the DEGs. The data obtained on the DEGs and their transcription factors allowed us to construct a map representing the relationships between the transcripts, the transcription factors, and the cellular processes in which they were involved upon treatment with CAP (Figure 4 and Figure 5). The genes were grouped into clusters based on their functional annotations (gene ontology (GO), Panther, and KEGG). It can be seen that the regulatory genes for the up-regulated sets were different for the A549 and Wi-38 cells. In addition, most of the cellular processes they up-regulated were the same for the A549 and Wi-38 (Appendix A). The activation of the p53 pathway, KRAS signaling, UV response, TNF-alpha signaling, and apoptosis-related processes occurred in both cell lines. The specific changes in the A549 tumor cells were the up-regulation of the groups of genes that respond to endoplasmic reticulum stress and the endoplasmic reticulum lumen.

In the case of the gene regulators of the down-regulated transcripts, there was a much greater similarity between the A549 and the Wi-38. Thus, the transcription factor SMAD4 is a specific down-regulator for A549 cells. Among the processes that are negatively regulated in Wi-38 cells is the “intrinsic apoptotic signaling pathway” (GO 0097193). It is possible that the suppression of the intrinsic apoptosis pathway may contribute to the better survival of Wi-38 cells after CAP treatment.

For the CAP-dependent down-regulated genes, the same transcription factors were identified for both cell lines: AR, E2F4, NFYB, NFYA.

For the A549 cells, the majority of the genes with reduced transcriptional activity were associated with the processes of cell division and cell cycling, including the group responsible for the G2/M transition. Nevertheless, some of the groups identified were the same as those found in the Wi-38 cells. It can be concluded that CAP irradiation suppresses mitosis predominantly in cancer cells. Among the down-regulated genes in Wi-38 cells, in addition to mitosis-related genes, there are genes whose products are involved in post-translational protein modification, through Golgi and ER membrane-related pathways.

### 3.2. Real-Time PCR Analysis of Gene Expression and Protein Level

Based on the transcriptome-analysis data, the genes *KLF4*, *FOS*, *ATF3*, and *GADD45B* were selected to confirm changes in their mRNA levels in response to CAP exposure between the two cell lines. The results of the RT-PCR assay showed (Figure 6a) that the *ATF3*, *FOS*, *GADD45B*, and *KLF4* mRNA levels were significantly different between the control and CAP-treated samples (*p* < 0.05) in the A549 cells. The expressions of the *KLF4*, *FOS*, and *ATF3* mRNAs in the A549 cells were up-regulated (by eight times and more) 3 h after the CAP treatment, and tended to return to baseline 24 h after irradiation. The same analysis of the Wi38 cells revealed weak changes in the expression of these genes. Thus, the RT-PCR data confirmed the data obtained from the transcriptome analysis of the CAP-exposed cells.

The Western blot analysis of the ATF3- and GADD45B-protein contents in the cell lysates of the control and CAP-treated cells showed that, indeed, the changes at the protein level corresponded to the changes detected at the mRNA level (Figure 6b).

The Western blot results showed that the ATF3 and GADD45B were significantly different between the control and experimental groups (*p* < 0.05). 

### 3.3. CAP Treatment Arrests the Cells in G2 Phase of Cell Cycle

To determine the cell-cycle changes identified from the transcriptome data, the semi-selective CAP regime was used for the treatment of the A549 cancer cells and the normal Wi-38 and MRC-5 cells. The samples were prepared from irradiated cells for cell-cycle analysis by flow cytometry. Colchicine was used for the treatment of the cells as a positive control of mitosis arrest. The data obtained show that the CAP exposure blocked the passing of the cancer cells from the G2 phase to cell division (M phase) in the cell cycle (Figure 7a,b). The increase in the proportion of A549 cells in the G2 phase from 15% in the control cells to about 60% occurred in the cells exposed by the CAP 30–60 s (Figure 7b). The Wi-38 and MRC-5 cells also showed an increase in the number of cells in the G2 phase after the CAP treatment, but this increase was less substantial than in the A549 cells. The pattern of disturbances in the cell cycle of the A549 upon CAP treatment was similar to that induced by colchicine (Figure 7c).

In order to understand whether a decrease in ROS would normalize the cell cycle after the CAP irradiation, the ROS inhibitor N-acetylcysteine (NAC) was added to the cells before the CAP treatment. The NAC is a synthetic derivative of the endogenous amino acid L-cysteine and a precursor of glutathione (GSH) [40]. It acts directly as a scavenger of free radicals, especially oxygen radicals [41]. We found that N-acetilcestine inhibits CAP-dependent cell-cycle arrest in cancer and normal cells (Figure 7b). This finding was in good agreement with the viability data—according to which the NAC suppressed the killing effects of the CAP (Figure 7e).

Next, we analyzed the increase in the proportion of the phosphorylated form of histone H2A.X (γH2AX) that can affect the success of the cell cycle [42]. The level of γH2AX was up-regulated for the cancer cells A549 and H23, as well as for the normal cells of Wi-38 (Figure 7f,g).

The MDM2 E3 ubiquitin ligase is a protein that is involved in the regulation of the p53 pathway. It targets wild-type p53 for degradation by the proteasome [43]. The up-regulation of *MDM2* was detected in the transcriptome of the Wi-38 cells (Figure 4b). Using the Western blot analysis, we found that the up-regulation of the MDM2 protein was an early response to the CAP treatment in the cancer and normal cells (Figure 7f,g).

Thus, we confirmed the transcriptome data on the activation of pathways associated with cell-cycle arrest and the p53 pathway.

### 3.4. Analysis of GPX4 and GPX7 Responses to the CAP Treatment

Since CAP stimulates RNS- and ROS-dependent cell death, we analyzed the change in glutathione peroxidases in the CAP-treated cells. Glutathione peroxidases protect cells against oxidative challenge, and they inhibit inflammation and oxidant-induced regulated cell death [24]. The A549 and H23 cancer cells and healthy Wi-38 cells were exposed to CAP under semi-selective conditions. Cell lysates were prepared 3 h and 24 h later for the analysis of the GPX4 and GPX7 by Western blot (Figure 8). We found that the A549 cells were GPX4-deficient cells, with a weak GPX4 response to CAP treatment (Figure 8a,b). The H23 cells demonstrated the same response to CAP treatment as the A549. The Wi-38 cells showed a strong basal level of GPX4, which was down-regulated by the CAP exposure. It can be hypothesized that high baseline GPx4 levels and/or their high functional activity in healthy Wi38 fibroblasts make it possible to overcome the lipid stress induced by CAP exposure.

The GPX7 is located in the endoplasmic reticulum. and it is a necessary enzyme involved in the oxidative folding of endoplasmic reticulum proteins [44]. The A549 cells presented the highest basal levels of GPX7. The GPX7 response to the CAP exposure was about the same for all three cell lines (Figure 8). The CAP treatment led to an increase in GPX7 three hours after the CAP treatment, with a slight decrease by 24 h. Thus, it can be hypothesized that CAP treatment slightly stimulates the oxidative protein’s folding in ER.

### 3.5. CAP Treatment Changes the Ultrastructure of Nucleolus in the A549 Cells

The cellular structures in the CAP-irradiated A549 cells’ ultrathin-section samples were analyzed. The most significant differences were observed in the ultrastructures of the ER lumens and nucleoli (Figure 9). The ER lumens expanded after 6 h of the CAP treatment and contained electron-lucent unstructured material (Figure 9a).

The nucleoli of the intact A549 cells (Figure 9b) were large and polymorphic, and they generally had medium electron density, with clear fibrillar centers. The electron-dense fibrillar component was poorly visualized. The granular component had medium electron density, with visible fibrils and small grains. The most prominent effect of the 6 h of exposure was a distinct increase in the density of electron-dense fibrillar and granular components of the nucleolus, with distinct cord formation. A day after the CAP exposure, the nucleoli retained the altered structure.

Thus, the CAP treatment of A549 cells leads to an increase in the electron density of the electron-dense fibrillar and granular components of the nucleoli, which may reflect an increase in the synthesis of ribosomal components.

### 3.6. Direct CAP Exposure of the Cultivated Cells Led to Changes in the Extracellular Contents of Nitrite Ions

It was demonstrated in our previous work [3] that a CAP treatment under a semi-selective regime induced the increase in intracellular ROS both in A549 cancer cells and in normal Wi-38 fibroblasts. Such increases were detected rapidly after the treatment, and, later, the ROS levels tended to revert to their initial values only in healthy Wi38 cells. Thus, healthy cells are better adapted to overcoming oxidative stress. Here we analyzed the ability of A549 and Wi38 cells to overcome the stress effects of CAP-dependent reactive nitrogen species.

The extracellular nitrite ion (NO_2_^−^) levels were estimated by a Griess reaction in a culture medium of cancer and normal cells treated with CAP under semi-selective conditions. 

The initial NO_2_^−^ levels were about the same for the cancer and normal cells (Figure 10, control curves), and remained unchanged without CAP exposure for the duration of the experiment. Thirty minutes after the CAP exposure, the NO_2_^−^ concentration rose, and the increase in the NO_2_^−^ concentration was apparently the same for the cancer and normal cells. Next, the increase in the NO_2_^−^ concentration was significantly higher for the tumor cells than for the healthy cells: the values were up to 165 µM and 70 µM, respectively (Figure 10, curves CAP 2 min). The longer exposure led to a greater increase in the concentration of extracellular NO_2_^−^.

Thus, the CAP treatment increased the extracellular nitrite levels in the tumor cells much more than in the healthy cells.

## 4. Discussion

Over the last decades, cold atmospheric-pressure plasma was widely studied for its antitumor potential, for the inactivation of pathogenic bacteria, bacteriophages and viruses [45,46,47], and for the functionalization of various surfaces [48,49]. In our study, semi-selective plasma-generation conditions were used: the CAP selectively killed A549 lung adenocarcinoma cells for 1 min of treatment, with a weak effect on the viability of healthy Wi-38 cells, as described previously [3]. We showed here that under these conditions, the nitrite-ion formation was higher in the extracellular medium of CAP-treated A549 tumor cells. Previously, we also found that the intracellular ROS was higher in A549 cells than in Wi-38 cells [3]. The differences between the ROS and RNS contents after the CAP treatment may have resulted from differences between the metabolism of tumor and healthy cells [23], and can regulate the viability of CAP-treated cells. These data are not new findings, as the critical importance of RNS and ROS generation for CAP-dependent cytotoxic activity has already been demonstrated [50]. Nevertheless, the detected differences in RNS and ROS accumulation confirm the semi-selectivity of the chosen conditions of CAP exposure.

Excess ROS and RNS, as well as UV radiation and ionizing radiation, lead to oxidative DNA damage. Such DNA damage can be indirectly detected by an increase in the proportion of the phosphorylated form of histone H2A.X (γH2AX) [42]. The DNA fragmentation in cells undergoing apoptosis also induces extensive H2A.X phosphorylation. The CAP treatment stimulated an increase in γH2AX in all the irradiated cells (Figure 7f,g). If there is little DNA damage and the reparation system is working correctly, such lesions can be repaired. However, in tumor cells, many vital systems function abnormally, and the excessive accumulation of DNA damage can significantly affect the cell cycle, especially in the S/G2/M phase. In this study, we showed a remarkable increase in the G2/M phase of the cell cycle in A549 after CAP treatment. This allows us to conclude that the G2/M block led to the persistent γH2AX foci observed 24 h after the CAP treatment. It should be noted that we irradiated the tumor and healthy cells at a time when the majority of all the cells were in G0/G1 phase. This is an important condition, since Keidar’s group suggested that the effect of CAP on cells depends on the number of cells within the population treated that are in the S-phase (i.e., based on differences in the distribution of cancer cells and normal cells within the cell cycle) [51]. However, the matching was not so clear-cut: their data showed that CAP affects all stages of the cell cycle; however, depending on the phase of the cell cycle in which a cell exists, the outcomes of the CAP treatment or the cell response are different. Our own data suggest that in normal cells, CAP treatment significantly disrupts the S/G2 transition.

A synthetic derivative of the endogenous amino acid L-cysteine and precursor of glutathione, N-acetylcysteine (NAC), is synthesized in all cells and protects them from oxidative stress [40]. It scavenges ROS through the reaction with its thiol group [41]. We found that NAC increased the cell viability of the CAP-treated cells and suppressed CAP-dependent cell-cycle arrest (Figure 7e). Our data confirm what was shown previously for human leukemia Molt-4 cells treated with helium plasma [52]. Thus, CAP-dependent cell-cycle arrest is a ROS/RNS dependent process.

Cellular lipids may also be targets of ROS [53]. Glutathione peroxidase 4 (GPx4) is the only enzyme in the GPX family that directly reduces and destroys lipid hydroperoxides [44]. It uses selenocysteine as the active center to catalyze the reduction in H_2_O_2_ or organic hydroperoxides, thereby reducing their toxicity and maintaining redox balance. Since the basal level of the GPX4 protein in A549 is extremely low (Figure 8), the accumulation of oxidized lipids in CAP-treated A549 cells may stimulate cell death. In contrast, GPX4 levels are high in Wi38, much higher than in H23 cells, and this probably allows the cells to metabolize oxidized lipids after CAP treatment and avoid death.

Examining the changes in mRNA in response to the CAP exposure, we concluded that the trends in the responses of the A549 cells and Wi-38 cells were the same, and this could indicate common groups of genes with similar responses. However, the amplitude of the response to the CAP was more variable in the A549 cells.

Currently, many datasets are already available on transcriptome changes in cells treated with CAP or a plasma-activated medium (PAM). The alterations in mRNA transcription in A549 cells in response to direct CAP treatment were already studied in a previous work [54]. The authors of that study found that the major effect of plasma exposure was the activation of MAPK and p53-signaling pathways, resulting in changes in the gene expression of MEKK, GADD, FOS, and JUN. The p53-regulated and DNA-damage-inducible protein, GADD45, plays a role in the G2/M checkpoint in response to DNA damage [55]. Furthermore, GADD45 up-regulation and p53-pathway activation were also found in A875 melanoma cells after direct CAP treatment [6]. Regarding cases of PAM exposure to tumor cells, DEG data can also be encountered both in the comparison of pre- and post-treatment and in the responses to treatment of healthy and tumor cells. For example, Jung C.-M. et al. treated thyroid cancer cell lines with PAM, and the gene-expression profiles were evaluated using RNA sequencing [56]. The authors showed that PAM induced the gene expression of the growth-arrest pathway and the GADD45a gene. They concluded that PAM stimulates cell death through the ROS-dependent activation of EGR1/GADD45a signaling. In our study we also detected GADD45a/b, EGR1, FOS, and JUN up-regulation (Figure 3) and the strong activation of the p53 pathway (Figure 4). It can be concluded that the activation of the EGR1/GADD45 cascade is one of the key transcriptional responses of tumor cells to direct exposure to CAP or a CAP-activated medium.

Oh C. et al. [57] analyzed changes in gene expression in head-and-neck cancer cells (HNC) and healthy fibroblasts treated with PAM. They showed that 199 genes were up-regulated and 213 genes were down-regulated in the HNC cells after PAM treatment, with no noticeable changes in gene expression in the healthy fibroblasts. They found that the deaths of the PAM-treated cancer cells were linked to excessive ROS accumulation in the mitochondria leading to the up-regulation of ATF4/CHOP activity. In our study, we also detected the up-regulation of activating transcription factor 4 and 3 (ATF4 and ATF3) mRNAs and the ATF3 protein. The ATF3 and ATF4 proteins are induced in response to endoplasmic reticulum (ER) stress or amino-acid starvation by a mechanism requiring the eIF2 kinases PERK (RNA-like endoplasmic reticulum kinase) [58,59].

A distinctive feature of the transcriptional responses of the A549 cells in our study is the activation of ER stress-related genes—a total of 18 genes from the up-regulated group, including ATF3 and ATF4 (Appendix A). In addition to the molecular changes after the CAP treatment, we also noted ultrastructural changes occurring in the cells. The changes in the ER-lumen size may reflect both normal metabolic changes associated with altered water–salt metabolism and ER stress. Since transcriptional changes in CAP-exposed cells indicate ER-stress-pathway activation, it can be hypothesized that lumen expansion is a macro-level response to such transcriptional changes. In support of the hypothesis that ER stress occurs is the translocation of calreticulin (CRT) from the ER to the outer plasma membrane of the cell, which we previously detected in CAP-treated cells [5,60]. A similar translocation of calreticulin has also been shown in other studies investigating the markers of immunogenic cell death induced by CAP treatment [61,62].

The changes in the nucleoli structure of the CAP-treated cells reflected an increase in the synthesis of ribosomal components. Nucleoli ensure the optimal efficiency of protein biosynthesis, they perform a key function in the maintenance of homeostasis in cells, and they can directly influence cell-cycle progression, cell growth, and proliferation [63,64]. The disruption of nucleolar morphology and/or functioning by any stressor can provoke the death of eukaryotic cells [65].

Taking into account the data obtained, we can assume the following mechanism of cell death under CAP exposure (Figure 11).

In the proposed mechanism, the pathways related to ER stress address only the effects in tumor cells. We hypothesize that the CAP-dependent activation of ER stress in A549 cells suggests their greater sensitivity to CAP compared to healthy Wi-38 cells.

## 5. Conclusions

The specific cytotoxic activity of CAP against tumor cells has a molecular basis. The treatment of healthy and tumor cells with CAP under conditions of selective tumor-cell death stimulates largely similar types of transcriptional response: the up-regulation of the p53 pathway, KRAS signaling, the UV response, TNF-alpha signaling, and apoptosis-related processes. Of great importance is the fact that the amplitude of the responses in tumor cells is much broader, and this can shift the equilibrium toward cell death. The greater sensitivity of tumor cells to CAP-induced ER stress also contributes to the stimulation of cell death.

It is notable that among the processes that are negatively regulated in Wi-38 cells is the “intrinsic apoptotic signaling pathway”. It is possible that the suppression of the intrinsic apoptosis pathway may contribute to the better survival of Wi-38 cells after CAP treatment. We have not yet found a sound molecular justification for the changes in the morphology of tumor-cell nucleoli after CAP irradiation. These data can be used as a basis for further studies, since little is known about the effect of CAP on the structure of nucleoli. To consider how general the patterns found are, it will be useful to perform similar studies on paired tumor and normal cells from other histological origins.

## Figures and Tables

**Figure 1 biomolecules-13-01672-f001:**
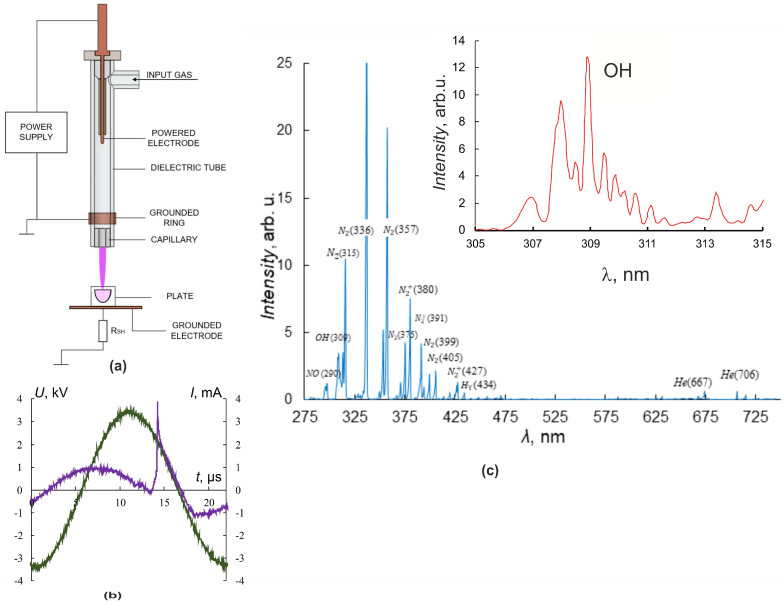
Plasma–source design and CAP jet characteristics. (**a**) Plasma–source design, (**b**) current (purple) and voltage (green) oscillograms. (**c**) CAP spectra in helium. The inset shows a part of the spectrum with peaks of OH.

**Figure 2 biomolecules-13-01672-f002:**
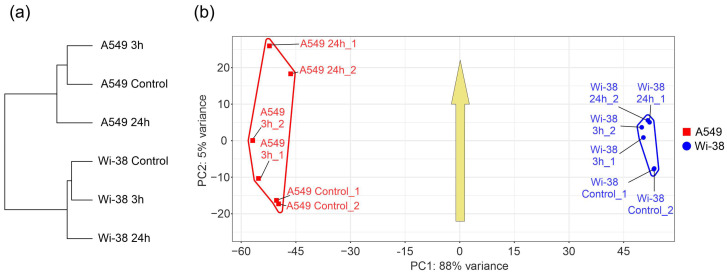
Relationships between individual transcriptomes and groups of transcriptomes of Wi-38 and A549 cells, treated with CAP for 1 min and cultivated for 3 h and 24 h. (**a**) A tree of Euclidean distances of variance stabilizing transformed (VST) RNA-expression data of Wi-38 and A549 cells. The complete agglomeration method was used for clustering. (**b**) Principal component analysis of DESeq2-normalized VST-transformed RNA-expression data. Sample-specific PC1:PC2 points are annotated with cell-defined envelopes. The yellow arrow shows the common trend of PC1:PC2 transition from non-treated (control) or treated with CAP and cultivated for 3 h and 24 h.

**Figure 3 biomolecules-13-01672-f003:**
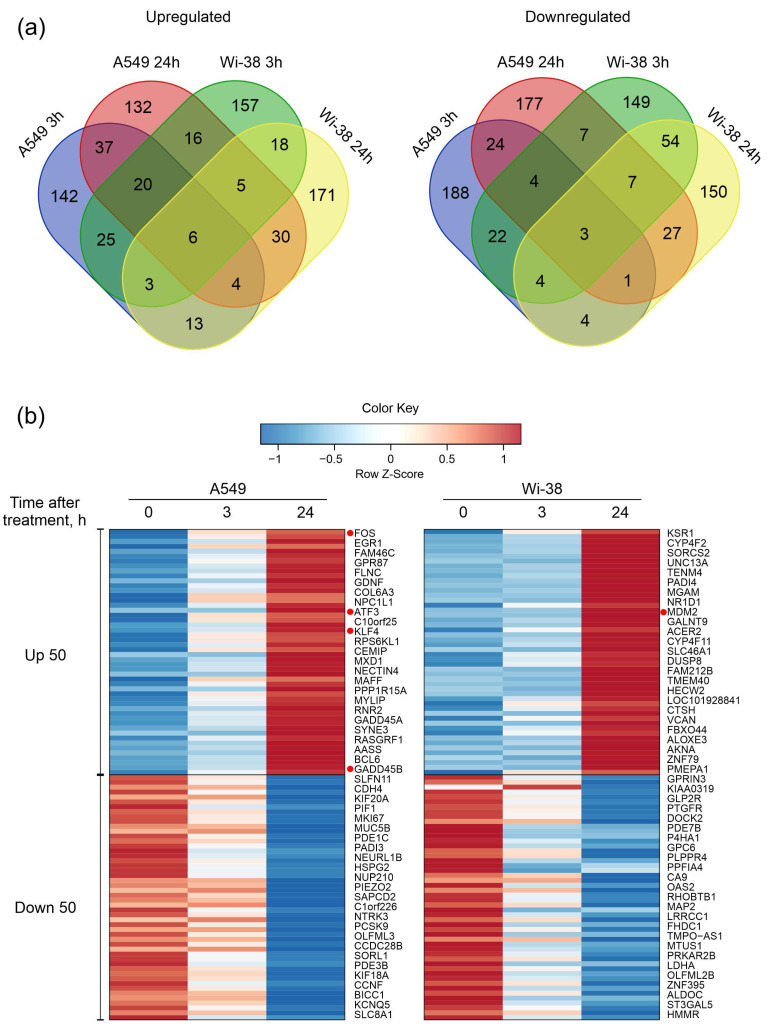
Characterization of differentially expressed gene (DEG) sets of A549 cancer cells and normal Wi-38 cells. (**a**) Venn diagrams showing intersections of DEGs sets, separately for RNAs with increased (Up-regulated) or decreased (Down-regulated) levels in CAP-treated compared to the corresponding non-treated cells. (**b**) Heatmap of the 50 most variable up-regulated and down-regulated differentially expressed genes in Wi38 and A549 cells. Transcripts discussed in the text are highlighted with red dots.

**Figure 4 biomolecules-13-01672-f004:**
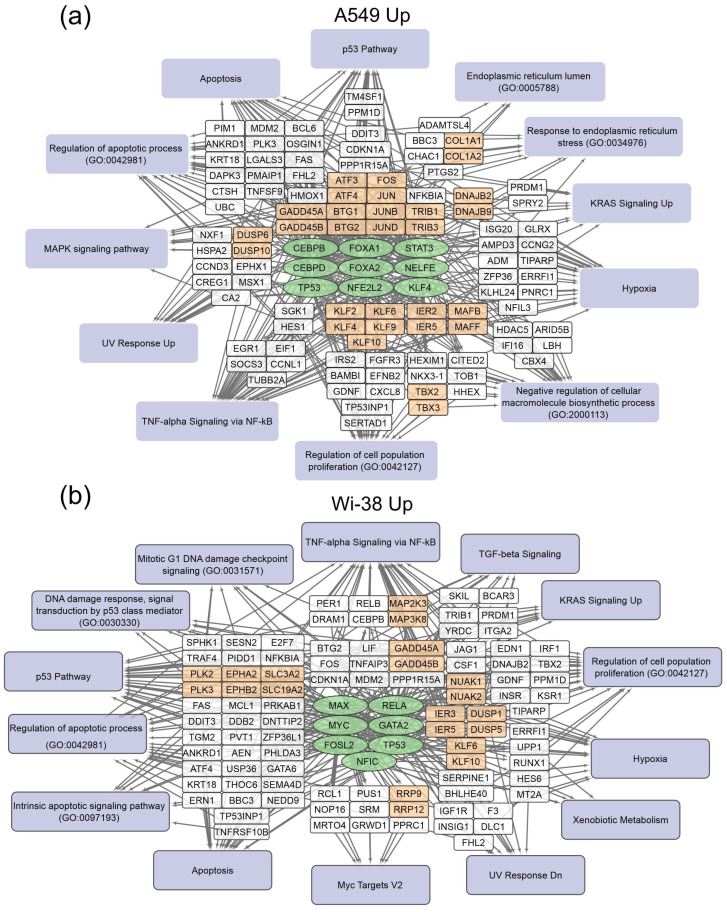
Schemes of the relationship between transcripts, transcription factors, and processes activated in A549 (**a**) and Wi-38 (**b**) cells upon treatment with CAP. Activated transcription factors are shown in green ovals; selected activated genes are in white and orange rectangles; signaling pathways, biological processes, and gene annotations are in lilac rectangles. Groups of genes with similar functions are drawn together. Based on the analysis of the top 250 activated genes using Enrichr libraries: “ENCODE and ChEA Consensus TFs from ChIP-X”; “MSigDB Hallmark 2020”; “GO Biologic Process 2021”; “Panther 2016”; and “KEGG 2021 Human”.

**Figure 5 biomolecules-13-01672-f005:**
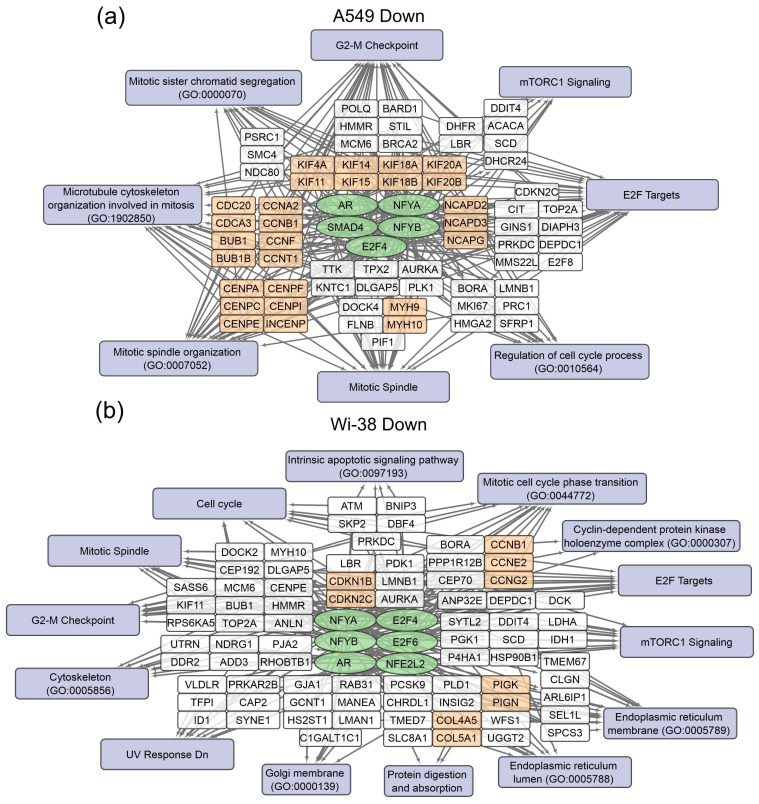
Schemes of the relationship between transcripts, transcription factors, and processes down-regulated in A549 (**a**) and Wi-38 (**b**) cells upon treatment with CAP. Down-regulated transcription factors are shown in green ovals; selected activated genes are in white and orange rectangles; signaling pathways, biological processes, and gene annotations are in lilac rectangles. Groups of genes with similar functions are drawn together. Based on the analysis of the top 250 activated genes using Enrichr libraries: “ENCODE and ChEA Consensus TFs from ChIP-X”; “MSigDB Hallmark 2020”; “GO Biologic Process 2021”; “Panther 2016”; and “KEGG 2021 Human”.

**Figure 6 biomolecules-13-01672-f006:**
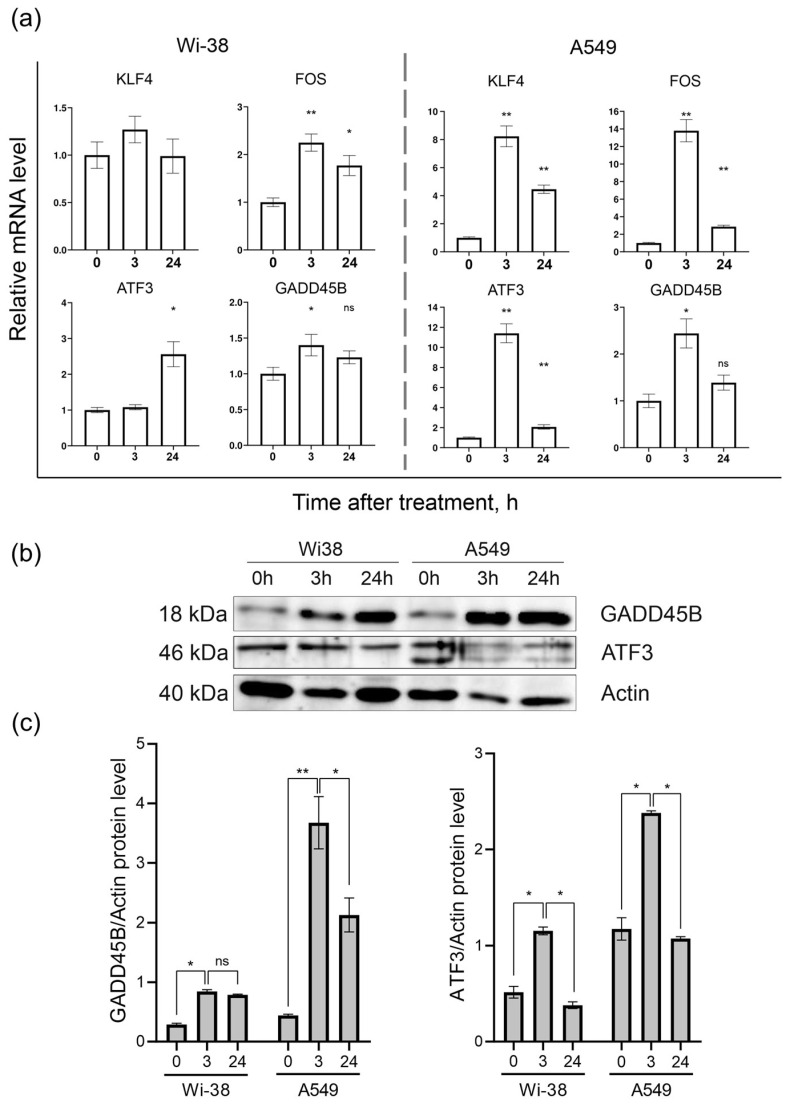
Analysis of mRNA levels of *KLF4*, *FOS*, *ATF3*, and *GADD45B* and protein levels of ATF3 and GADD45B in CAP-exposed cells (helium plasma jet, 3.3 kV *f_U_* = 52 kHz). (**a**) RT-PCR data and (**b**) Western blot data. (**c**) The quantification of protein band intensities from two independent Western blot experiments. The original Western blot images of (**b**) can be found in Appendix A. * *p* < 0.05, ** *p* < 0.005, ns—non significant.

**Figure 7 biomolecules-13-01672-f007:**
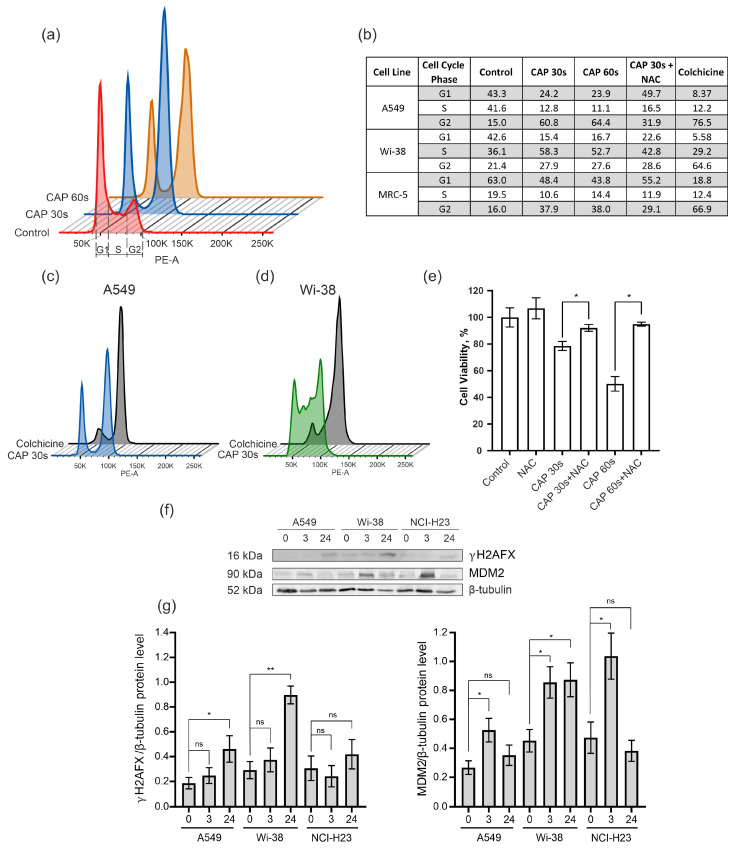
Effect of CAP treatment on the cell cycle. (**a**–**e**) The A549, Wi-38, and MRC-5 cells were treated with CAP for 30 s and 60 s (helium plasma jet, 3.3 kV *f_U_* = 52 kHz); 24 h later, cells were fixed and stained, as described in *Methods*. Control—untreated cells. (**a**) An example analysis of A549 cells with G1, S, and G2. (**b**) The distribution of cells in a particular phase of the cell cycle. (**c**,**d**) Example of analysis of CAP-treated A549 cells and Wi-38 cells compared to colchicine treatment. (**e**) The influence of NAC on viability of A549 cells 24 h after the CAP treatment. (**f**) Representative image of Western blot analysis of A549, Wi-38, and H23 cells and (**g**) quantifications of the signals corresponding to γH2A.X and MDM2. Control—untreated cells. Data are presented as average value ± SD. The differences are significant with * *p* < 0.05 and ** *p* < 0.005 between two groups; ns—non-significant. The original Western blot images of (**f**) can be found in Appendix A.

**Figure 8 biomolecules-13-01672-f008:**
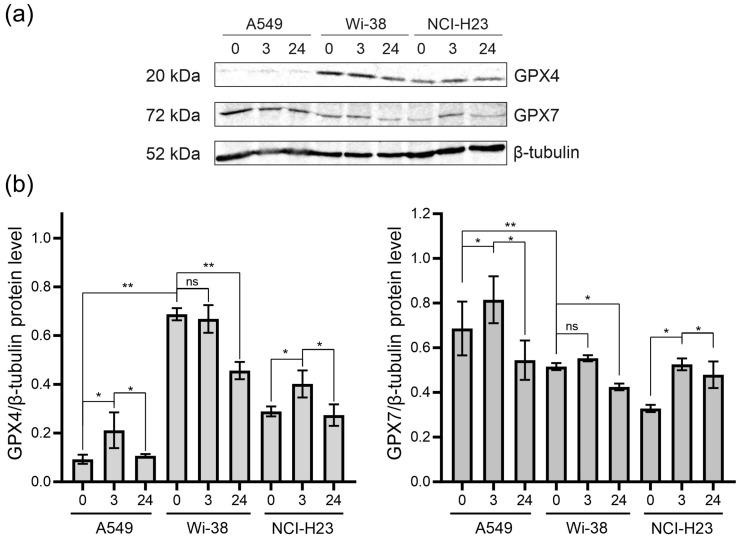
The GPX4 and GPX7 responses to the CAP treatment. The A549, Wi-38 and H23 cells were treated with CAP for 1 min (helium plasma jet, 3.3 kV *f_U_* = 52 kHz); 3 h and 24 h later, cell lysates were prepared and GPX4 and GPX7 were analyzed by Western blot, as described in Methods. Representative image of Western blot analysis (**a**) and quantifications of the signals corresponding to GPX4 (**b**) and GPX7. Control—untreated cells. The quantification of protein band intensities from two independent Western blot experiments was performed with Image Lab5.1 (BioRad). The signal intensity was normalized to the loading-control signal (tubulin). Data are presented as average value ± SD. The differences are significant with * *p* < 0.05 and ** *p* < 0.005 between two groups; ns—non-significant. The original Western blot images of (**a**) can be found in Appendix A.

**Figure 9 biomolecules-13-01672-f009:**
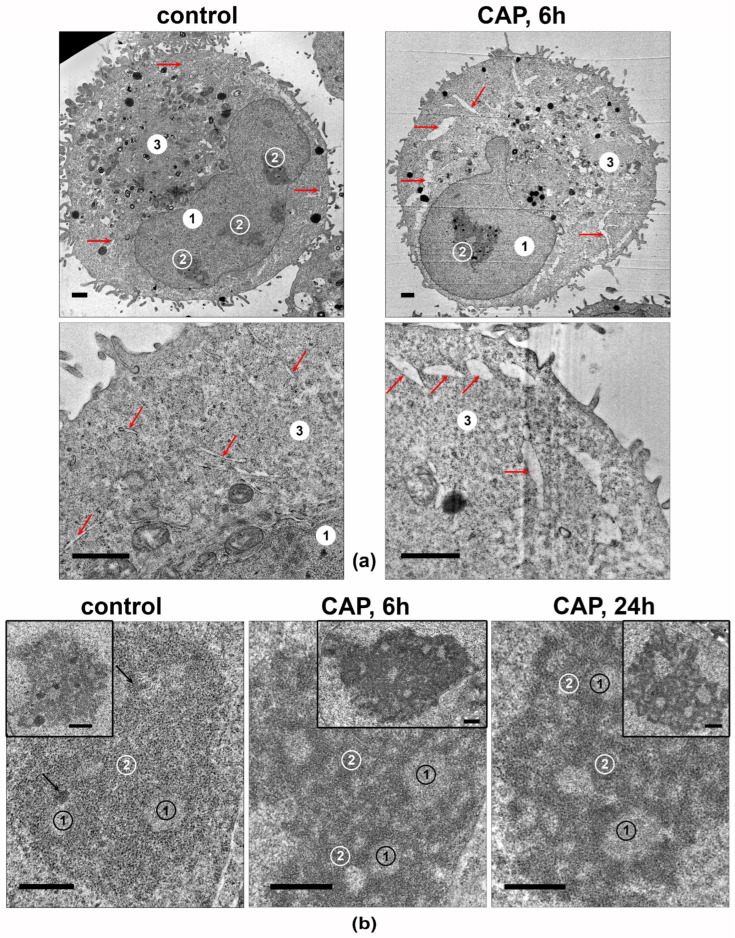
Transmission-electron micrographs showing the ultrastructures of A549 cells. Samples of control and CAP-exposed cells (helium plasma jet, 3.3 kV *f_U_* = 50 kHz). (**a**) Representative images of cells and their fragments. 1—nucleus, 2—nucleolus, 3—cytoplasm; arrows indicate ER lumens. (**b**) Representative images of nucleoli in CAP-treated A549 cells. The inserts show the full view of the nucleoli. 1—fibrillar center. 2—granular component; arrows indicate electron-dense fibrillar component. Ultrathin sections with scale bar = 1 µm (**a**) and scale bar = 500 nm (**b**).

**Figure 10 biomolecules-13-01672-f010:**
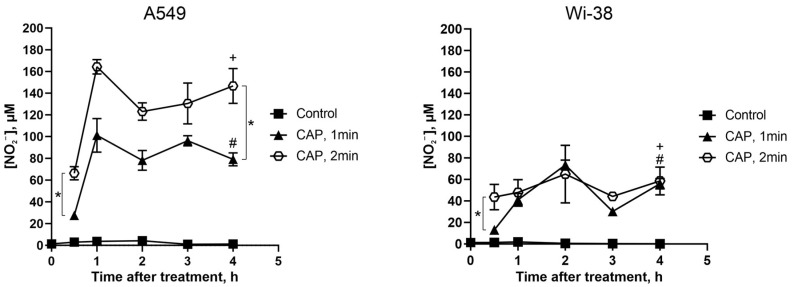
Direct CAP treatment of the growing cells increases the extracellular nitrite-ion levels. The A549 and Wi-38 cells were treated with CAP for 1 min and 2 min (helium plasma jet, 3.3 kV *f_U_* = 52 kHz); 30 min–4 h later, culture medium was collected and analyzed in Griess reaction by spectrophotometric method (λ = 570 nm). Control—culture medium of growing cells without treatment. Data are presented as average value of four technical repeats ± SD. The differences are significant with * *p* < 0.05 between two groups. (+) and (#) *p* < 0.05 between the same two groups in A549 and Wi-38 cells.

**Figure 11 biomolecules-13-01672-f011:**
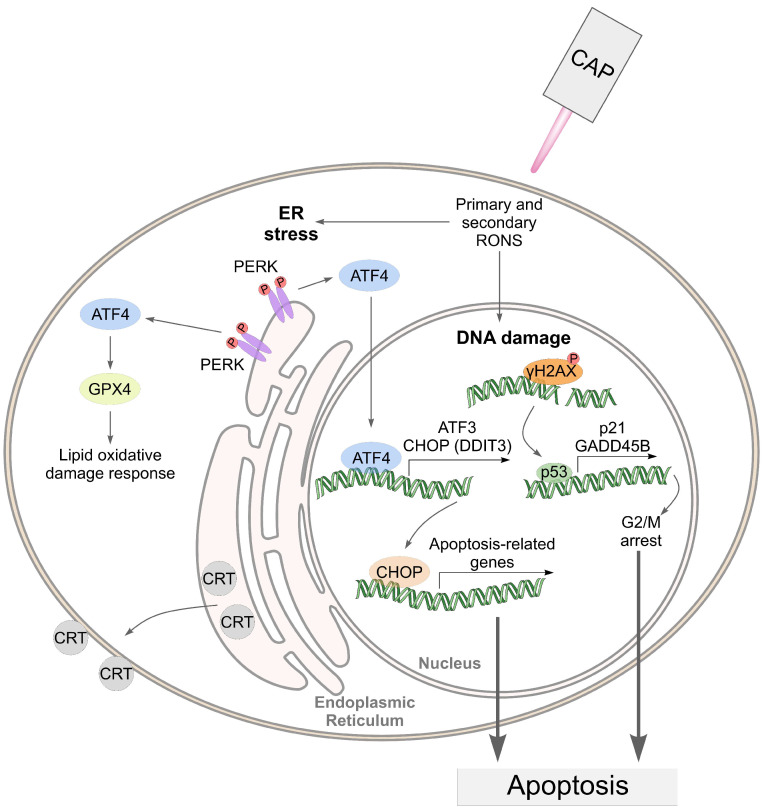
An illustration of the death-related pathways that can be activated by cold atmospheric plasma (CAP) in cancer cells.

**Table 1 biomolecules-13-01672-t001:** Primers for RT-qPCR.

Gene	Sequence 5′ → 3′	Product Size, bp
*HPRT1*	F: CTCGAGATGTGATGAAGGAG	258
	R: TATCTTCCACAATCAAGACATT	
*GAPDH*	F: GAAGATGGTGATGGGATTTC	226
	R: 5′-GAAGGTGAAGGTCGGAGT	
*KLF4*	F: GGGAGAAGACACTGCGTCAA	229
	R: TCCAGGTCCAGGAGATCGTT	
*FOS*	F: GCGTTGTGAAGACCATGACAG	165
	R: GTGTATCAGTCAGCTCCCTCC	
*ATF3*	F: TTTGCTAACCTGACGCCCTT	218
	R: TTGTTTCGGCACTTTGCAGC	
*GADD45B*	F: GTACGAGTCGGCCAAGTTGA	263
	R: CCGTGTGAGGGTTCGTGAC	
*GPX4*	F: GTGAGGCAAGACCGAAGTA	275
	R: TCCACTTGATGGCATTTCCC	
*GPX7*	F: ACTTCAAGGCGGTCAACATC	235
	R: CGGGCAAAGCTCTCAATCTC	

**Table 2 biomolecules-13-01672-t002:** Common transcription factors involved in the response of A549 cells to CAP treatment. The table includes selected results of the Enrichr analysis of the top 250 transcripts that are up-regulated (Up) or down-regulated (Down) 3 h and 24 h after CAP treatment (Enrichr terms of “ENCODE and ChEA Consensus TFs from ChIP-X” library).

Type of Regulation	Selected Terms	Representative Genes	Number of Genes
Up	CEBPB ENCODE	*CBX4; GADD45A; SPRY4; ARID5B; KLHL24; FBXO32; ISG20; BCL6; NFIL3; DDIT3*	23
	CEBPD ENCODE	*PPP1R15A; HDAC5; BTG2; BTG1; TOB1; FAM117A; HHEX; NFIL3; JUNB; IER5*	58
	FOXA1 ENCODE	*HEXIM1; PPP1R15A; CHIC2; BTG1; CXCL8; CITED2; FOS; DUSP6; BCL6; KLF9*	29
	FOXA2 ENCODE	*HEXIM1; PPP1R15A; ERRFI1; CITED2; PTPRH; TM4SF1; DUSP6; DUSP10; KLF5; TP53INP2*	36
	KLF4 CHEA	*ZNF296; CDKN1A; CITED2; ZFAND2A; ADM;GATA2; HOXB13; EFNB2; SOCS3; ZFP36*	61
	NELFE ENCODE	*HEXIM1; EGR1; MYLIP; BTG1; GADD45B; GADD45A; CITED2; FAM46C; GDF15; FOS*	38
	NFE2L2 CHEA	*CHIC2; CDKN1A; BTG1; CITED2; IRS2; PTGS2; PPM1D; RND3; OASL; EFNB2*	70
	STAT3 ENCODE	*EGR1; JUN; BTG2; BTG1; GADD45A; FOS; PRDM1; TOB1; PNRC1; ISG20*	43
	TP53 CHEA	*CDKN1A; BTG2; BTG1; PRDM1; ADRB2; PPM1D; BBC3; SERTAD1; PRX; SESN2*	38
Down	AR CHEA	*FBN2; USP13; BNC2; PRKDC; LAMA3; ADAMTS12; SPATA5; LPP; PTPRF; SLC8A1*	67
	E2F4 ENCODE	*TOP2A; ARHGAP11A; CCNF; KIF14; HJURP; BUB1B; KIF11; TTF2; MKI67; SMC4*	79
	NFYA ENCODE	*TOP2A;CCNF;KIF14;HJURP;TTF2;CDC20;TNFAIP8L1;GTSE1;LBR;DLGAP5*	97
	NFYB ENCODE	*TOP2A;GLDC;CCNF;KIF14;HJURP;KIF11;TTF2;MKI67;LMNB1;CDC20*	145
	SMAD4 CHEA	*PRELID2; NRP2; PKDCC; BNC2; LYPD1; PRICKLE2; NRXN3; ARHGAP26; PLEKHA7; SLC8A1*	41

**Table 3 biomolecules-13-01672-t003:** Common transcription factors involved in the responses of Wi-38 cells to CAP treatment. The table includes selected results of the Enrichr analysis of the top 250 transcripts that are up-regulated (Up) or down-regulated (Down) 3 h and 24 h after CAP treatment (Enrichr terms of “ENCODE and ChEA Consensus TFs from ChIP-X” library).

Type of Regulation	Selected Terms	Representative Genes	Number of Genes
Up	FOSL2 ENCODE	*GADD45A; ITGA2; TNFAIP3; NR1D1; DUSP14; ITPKC; BFSP1; DDIT3; ZNF219; ZBTB7B*	30
	GATA2 CHEA	*BTG2; CDKN1A; TENM4; FHL2; KLHDC7A; CLN8; PPCDC; ZFP36L2; SLC9A1; FAM110A*	55
	MAX ENCODE	*HES6; PLK3; DYRK3; PANK1; DUSP1; INSR; SPHK1; PVT1; THUMPD2; TNFAIP3*	81
	MYC ENCODE	*PLK3; DUSP1; PVT1; THUMPD2; AEN; SLC3A2; NR1D1; ZBTB5; SNHG5; ZFP36L2*	58
	NFIC ENCODE	*TRAF4; GADD45A; DUSP1; ZFAS1; BHLHE40; MDM2; ZBTB7B; SLC25A45; PPM1D; DDB2*	27
	RELA ENCODE	*CSF1; KSR1; ZFAS1; TNFAIP3; SLC3A2; ZFP36L2; FAM110A; ERN1; ACTA2; DNAJB2*	38
	TP53 CHEA	*BTG2; CDKN1A; DYRK3; CEP85L; FHL2; ADRB2; PPM1D; EDA2R; BBC3; PIDD1*	47
Down	AR CHEA	*USP13; INSIG2; LRRC17; CHRDL1; SLC8A1; RHOBTB1; DTWD2; ADAMTS15; SH3BGRL; LEPR*	56
	E2F4 ENCODE	*TOP2A; CDKN2C; BORA; FANCL; IFT80; KIF11; CTDSPL2; DCK; LMNB1; AURKA*	37
	E2F6 ENCODE	*HIBADH; CCDC126; SPIN4; CETN3; VLDLR; MANEA; LMNB1; PDK3; SLC25A40; FAM3C*	106
	NFE2L2 CHEA	*INSIG2; GCNT1; PPM1H; ADD3; TFPI; SLC4A4; NALCN; TMTC4; GJA1; PRKAR2B*	59
	NFYA ENCODE	*TOP2A; KDM3A; ARL6IP1; HIBADH; PIGN; DCUN1D4; CETN3; SEL1L; HMMR; TMTC4*	81
	NFYB ENCODE	*TOP2A; ARL6IP1; HIBADH; PIGN; CETN3; KIF11; CTDSPL2; NXT2; LMNB1; HS2ST1*	124

## Data Availability

The RNA-Seq data were deposited at the NCBI SRA database (SRA Accession PRJNA1040866, https://www.ncbi.nlm.nih.gov/sra/PRJNA1040866). All data used to support the findings of this study are available from the corresponding author upon request.

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
