# Peer review of "The Molecular Basis for Selectivity of the Cytotoxic Response of Lung Adenocarcinoma Cells to Cold Atmospheric Plasma"

_biomolecules, 2023, doi:10.3390/biom13111672_

Round 1
Reviewer 1 Report
Comments and Suggestions for Authors
This paper presents the molecular basis for selectivity of the cytotoxic response of lung adenocarcinoma cells and lung fibroblasts to cold atmospheric plasma. The authors investigated the transcriptome analysis, assessed gene expression and protein levels in A549 and Wi-38 cells following CAP treatment. The authors also presented the endoplasmatic reticulum stress and ER lumens using the molecular methods and electron microscopy. The paper is well structured and add new insights into the mechanism of semi-selective regime for the direct CAP jet treatment of cancer cells. I recommend this paper to be published in the Biomolecules with minor revision.
1. Did the authors evaluate the thermal effects of the plasma jet or the temperature of CAP treated medium?
2. Apoptosis is closely related to cytotoxicity and cell cycle modulation. On Page 3 the authors indicated the cell cycle phase distribution. It would be of advantage if the authors also included data on cell apoptosis to further enhance the understanding of the semi-selective regime cytotoxicity.
3. Page 15 line 389, it was observed that A549 cells presented the highest basal level of GPX7. Please include a discussion on the impact of GPX7 on oxidative stress in cells.
4. Page 16 figure 8: all the pictures should add the scale bars for accurate measurement.
Author Response
This paper presents the molecular basis for selectivity of the cytotoxic response of lung adenocarcinoma cells and lung fibroblasts to cold atmospheric plasma. The authors investigated the transcriptome analysis, assessed gene expression and protein levels in A549 and Wi-38 cells following CAP treatment. The authors also presented the endoplasmatic reticulum stress and ER lumens using the molecular methods and electron microscopy. The paper is well structured and add new insights into the mechanism of semi-selective regime for the direct CAP jet treatment of cancer cells. I recommend this paper to be published in the Biomolecules with minor revision.
Answer
We thank the reviewer for appreciation of our work.
Comment 1. Did the authors evaluate the thermal effects of the plasma jet or the temperature of CAP treated medium?
Answer
The temperature measurement was added to the Methods section (Please, see L. 136-144).
Comment 2. Apoptosis is closely related to cytotoxicity and cell cycle modulation. On Page 3 the authors indicated the cell cycle phase distribution. It would be of advantage if the authors also included data on cell apoptosis to further enhance the understanding of the semi-selective regime cytotoxicity.
Answer
We thank the reviewer for this suggestion. We have already presented such data (for the same treatment conditions and cell lines) in our previous paper. Here, we added clarification regarding the type of cell death observed (L. 80-81): Using pan-caspase inhibitor Z-vad we revealed that the caspase-dependent cell death pathways made a major contribution to CAP-dependent cell killing.
Comment 3. Page 15 line 389, it was observed that A549 cells presented the highest basal level of GPX7. Please include a discussion on the impact of GPX7 on oxidative stress in cells.
Answer
Here, we included the addition (L. 422-423 and L426-427): “GPX7 is located in the endoplasmic reticulum and is necessary enzyme involved in the oxidative folding of endoplasmic reticulum proteins [44]. Thus, it can be hypothesized that CAP treatment slightly stimulates the oxidative protein’s folding in ER”.
Comment 4. Page 16 figure 8: all the pictures should add the scale bars for accurate measurement.
Answer
Scale bars were added to the Figure 9 (former Fig. 8).

Reviewer 2 Report
Comments and Suggestions for Authors
The manuscript entitled “The molecular basis for selectivity of the cytotoxic response of lung adenocarcinoma cells to cold atmospheric plasma” by Biryukov et al. investigated the responses to nuclear stress and ER stress constitute the main differences in the sensitivity of tumor and healthy cells to CAP exposure. This manuscript is moderately fine at this stage and needs to include many technical and scientific parts and the comments are listed below.
1. The author should also present schematic of device showing electrode configurations, used in the study in initial figures.
2. The author must include physical properties including OES and I/V of plasma used in the study. Plasma properties vary time by time; therefore, it must be presented to validate biological effect. The author should include voltage current profile, plasma gas temperature, OES, RONS profile, Conductivity, ORP, and other physical properties during variable exposures. Even though this kind of device reported previously, author must do it again to validate their findings.
3. The author must include scale bar in Fig 8. Either it is very weak or absent.
4. Author should validate overall phenomenon or mechanism using proper ROS scavengers or inhibitors.
5. The author must make a final concluding mechanism graphic as last figure of this manuscript.
Comments on the Quality of English LanguageEnglish language is moderately fine, minor spell check required.
Author Response
Dear Editors,
Please, let the reviewers know that we are very grateful for their productive comments on our work. We improve the article as was requested by the reviewers. Two new figures have been added as requested by the reviewers (Figure 1 and Figure 11).
Reviewer 2
The manuscript entitled “The molecular basis for selectivity of the cytotoxic response of lung adenocarcinoma cells to cold atmospheric plasma” by Biryukov et al. investigated the responses to nuclear stress and ER stress constitute the main differences in the sensitivity of tumor and healthy cells to CAP exposure. This manuscript is moderately fine at this stage and needs to include many technical and scientific parts and the comments are listed below.
Comment 1. The author should also present schematic of device showing electrode configurations, used in the study in initial figures.
Answer
Figure 1 with electrode configurations was added to the manuscript.
Comment 2. The author must include physical properties including OES and I/V of plasma used in the study. Plasma properties vary time by time; therefore, it must be presented to validate biological effect. The author should include voltage current profile, plasma gas temperature, OES, RONS profile, Conductivity, ORP, and other physical properties during variable exposures. Even though this kind of device reported previously, author must do it again to validate their findings.
Answer
Relevant data have been added to the text (Please, see L. 111-118 and L. 126-135) and to Figure 1.
Comment 3. The author must include scale bar in Fig 8. Either it is very weak or absent.
Answer
Scale bars were added to the Figure 9 (former Fig. 8).
Comment 4. Author should validate overall phenomenon or mechanism using proper ROS scavengers or inhibitors.
Answer
We used N-Acetyl-D-cysteine (NAC) in our work to validate ROS-dependent cell death phenomenon (please, see Fig. 7 b, e). We added a specification in the text that NAC is a ROS scavenger (please, see L383-384 and L. 524-525).
Comment 5. The author must make a final concluding mechanism graphic as last figure of this manuscript.
Answer
Many thanks to the reviewer for this valuable suggestion. We have added Figure 11 with the concluding molecular mechanism of CAP action and the text for this figure (Please, see L. 588-598).

Round 2
Reviewer 2 Report
Comments and Suggestions for Authors
I recommend accepting this manuscript as the author already revised it.
Comments on the Quality of English LanguageI recommend accepting this manuscript as the author already revised it.